# Adenosine and Its Receptors: An Expected Tool for the Diagnosis and Treatment of Coronary Artery and Ischemic Heart Diseases

**DOI:** 10.3390/ijms21155321

**Published:** 2020-07-27

**Authors:** Marine Gaudry, Donato Vairo, Marion Marlinge, Melanie Gaubert, Claire Guiol, Giovanna Mottola, Vlad Gariboldi, Pierre Deharo, Stéphane Sadrin, Jean Michel Maixent, Emmanuel Fenouillet, Jean Ruf, Regis Guieu, Franck Paganelli

**Affiliations:** 1Department of Vascular Surgery, Timone Hospital, F-13008 Marseille, France; marine.gaudry@ap-hm.fr; 2C2VN, INSERM, INRA, Aix-Marseille University, F-13015 Marseille, France; Donato.vairo@univ-amu.fr (D.V.); marion.marlinge@ap-hm.fr (M.M.); melanie.gaubert@ap-hm.fr (M.G.); claire.guiol@univ-amu.fr (C.G.); Giovanna.motto@ap-hm.fr (G.M.); Vlad.gariboldi@ap-hm.fr (V.G.); pierre.deharo@ap-hm.fr (P.D.); Emmanuel.fenouillet@univ-amu.fr (E.F.); jean.ruf@sfr.fr (J.R.); guieu.regis@orange.fr (R.G.); 3Laboratory of Biochemistry, Timone Hospital, F-13008 Marseille, France; 4Department of Cardiac Surgery, Timone Hospital, F-13008 Marseille, France; 5Department of Cardiology, Timone Hospital, F-13008 Marseille, France; 6Laboratoire Denel-Codifra, F-78150 Le Chesnay, France; stephane.sadrin@gmail.com; 7Unité de Recherche Clinique Pierre Deniker (URC C.S. 10587) Centre Hospitalier Henri Laborit, 86000 Poitiers, France; 8I.A.P.S. Equipe Emergeante, Université de Toulon, 83957 Toulon-La Garde, UFR S.F.A., F-86073 Poitiers, France; 9Department of Cardiology, Nord Hospital, ARCHANTEC, F-13015 Marseille, France

**Keywords:** adenosine, purinergic system, coronary artery disease, A_2A_ receptor, heart

## Abstract

Adenosine is an endogenous nucleoside which strongly impacts the cardiovascular system. Adenosine is released mostly by endothelial cells and myocytes during ischemia or hypoxia and greatly regulates the cardiovascular system via four specific G-protein-coupled receptors named A_1_R, A_2A_R, A_2B_R, and A_3_R. Among them, A_2_ subtypes are strongly expressed in coronary tissues, and their activation increases coronary blood flow via the production of cAMP in smooth muscle cells. A_2A_ receptor modulators are an opportunity for intense research by the pharmaceutical industry to develop new cardiovascular therapies. Most innovative therapies are mediated by the modulation of adenosine release and/or the activation of the A_2A_ receptor subtypes. This review aims to focus on the specific exploration of the adenosine plasma level and its relationship with the A_2A_ receptor, which seems a promising biomarker for a diagnostic and/or a therapeutic tool for the screening and management of coronary artery disease. Finally, a recent class of selective adenosine receptor ligands has emerged, and A_2A_ receptor agonists/antagonists are useful tools to improve the management of patients suffering from coronary artery disease.

## 1. Introduction

Adenosine, a nucleoside that is an ATP derivative, is found in most vascular beds of mammalian species. The first work in favor of the involvement of adenosine in the heart was that of Drury and Szent-Györgyi [1], and its contribution to coronary vasodilation was established by Berne [2]. Adenosine is currently used in the cardiology area for diagnostic and sometimes therapeutic use, particularly in coronary artery disease. The effects of adenosine on coronary blood flow are thought to be mediated primarily by the activation of A_2_ receptors [3,4,5], and mostly via the A_2A_ R subtype [6]. This vasorelaxation decreases vascular resistance, thereby facilitating coronary blood flow and oxygen delivery. This article reviews the most recent advances in emerging data on circulating plasma adenosine and its relationship with the A_2A_ receptor in coronary artery and ischemic heart diseases.

## 2. Metabolism of Adenosine

Adenosine is ubiquitously present, and is both an intermediate metabolite and a neuromodulator. Adenosine comes mainly from adenine nucleotide dephosphorylation at both the extracellular and intracellular levels. Adenosine can be generated at the intracellular space by AMP or S-adenosyl-homocysteine hydrolysis via cytosolic nucleotidase and S-adenosyl-homocysteine hydrolase, respectively (Figure 1). At the extracellular level, adenosine comes from the dephosphorylation of nucleotides (i.e., ATP, ADP, and AMP) mainly through the activation of ectonucleotidases CD39 and CD73. Adenosine is released in the extracellular spaces mostly via equilibrative nucleoside transporters (ENT-1) [7]. In the extracellular spaces, the behavior of adenosine is three-fold: (i) it is uptaken by red blood cells via ENT-1. (ii) Adenosine is transformed into inosine by adenosine deaminase (ADA; Figure 1). ADA is expressed particularly in red blood cells and at the mononuclear cell surface. In this last case, adenosine deaminase binds to CD26 and prevents the accumulation of adenosine, which is toxic for lymphocytes [8]. After deamination into inosine, adenosine joins the uric acid metabolism; thus, there is a link between adenosine and uric acid [9]. (iii) Finally, the part of adenosine that has not been taken up or deaminated interacts through binding to its receptors, leading to a strong impact on the cardiovascular system. At the extracellular level, adenosine impacts the cardiovascular system through the activation of four distinct G-protein-coupled receptor subtypes, denoted A_1_R, A_2A_R, A_2B_R, and A_3_R [10,11].

## 3. Adenosine Receptor Expression, Homocysteine, and Uric Acid

Extracellularly, adenosine activates four distinct receptors named A_1_R, A_2A_R, A_2B_R, and A_3_R, depending on their primary sequence and pharmacological properties [10,11,12,13].

A_1_ and A_3_R are coupled with Gi proteins, which inhibit the adenylyl cyclase and lower intracellular levels of cAMP. In contrast, A_2A_R and A_2B_R activate Gs proteins, inducing an increase in the cAMP level in target cells [10,11,12,13]. A_1_R and A_2A_R have a high affinity for adenosine, while A_2B_R and A_3_R have a lower affinity [10,11,13]. The activation of A_1_R and A_3_R leads to protection against myocardium/reperfusion injuries [14,15,16,17]. The activation of A_1_R protects against oxidative stress during the experimental myocardial ischemia-reperfusion process [16]. A_2A_R is strongly expressed in myocytes in humans, and its activation leads to the myorelaxation of smooth muscle cells and vasodilation, increasing the coronary blood flow [18] partly via cAMP production in smooth muscle cells, cAMP production and coronary vasodilation being correlated [19]. Vasodilation occurs through the activation of potassium channels (K_ATP_, Kv) and via the inhibition of calcium currents [3,4,5] (Figure 2).

Because adenosine strongly impacts the cardiovascular system via its receptors, the expression level and their functional activity are, therefore, of importance. Endogenous adenosine will be more potent when the adenosine receptor subtypes are highly expressed than when they are less abundant. Beside expression, receptor sensitivity plays also a major role and may be different from one tissue to another. The contribution of A_2A_R and A_2B_R to coronary vasodilation is particularly marked in the adaptive response to a decrease in O_2_ delivery and thus during myocardial ischemia [20]. The expression level of A_2_R subtypes varies according to the animal species. All subtypes are expressed in the smooth muscle and endothelial cells of the coronary arteries in most mammals, including rats, guinea pigs, hamsters, mice, dogs, pigs, and humans [20]. However, their expression level may differ from one tissue to another [21,22]. Thus, in a swine model a more abundant A_2B_R vs. A_2A_R expression in was shown in coronary vessels, despite a prominent function for A_2A_R vs. A_2B_R [6].

Although A_2A_R is strongly expressed in coronary arteries, it is difficult to investigate the adenosinergic pathway in the coronary arteries. The fact that the expression and function of A_2A_R in heart tissues [23] and coronary arteries [24] correlate with the expression and function of A_2A_R in peripheral blood mononuclear cells provides a unique window to link the adenosinergic system to ischemia in the coronary arteries. Previously, Adonis is an antibody directed to the human A_2A_R, that exhibited agonist properties [25]. These agonist properties were assessed by cAMP production, which leads to coronary vasodilatation. Thus, using Adonis it can be possible to simultaneously evaluate the A_2A_R expression (related to the presence of coronary stenosis) and functional response (by cAMP release) mainly related to the presence of myocardial ischemia [26]. The A_2A_R from patients with coronary artery disease (CAD) is poorly expressed and, consequently, their activation leads to the production of a low intracellular concentration of cAMP. These two characteristics have been shown to be associated with myocardial ischemia, as documented by positive exercise stress testing [26]. Because part of adenosine comes from the methionine cycle, there is a correlation between adenosine and the homocysteine plasma levels [27]. Furthermore, uric acid is the final product of adenosine degradation, which explains why adenosine and the uric acid plasma levels are correlated in CAD patients [27]. Interestingly, uric acid is significantly associated with CAD [28] and endothelial dysfunctions [9]. Thus, a high uric acid plasma level may be partly linked to adenosine and homocysteine metabolisms.

## 4. Modulating the Adenosinergic System: An Expected Tool for the Detection of CAD Patients

Exercise stress imaging has increasingly been employed for the detection of CAD. The underlying principle of the exercise test is that when the myocardium is under exercise conditions, the diseased ventricle receives less blood flow than the normal heart muscle. Thus, in CAD patients image abnormalities result from the heterogeneity of the coronary blood flow reserve between the normal and ischaemic areas. In many cases, patients cannot perform exercise, and pharmacological stressors might be a useful tool. Pharmacological stressors induced coronary vasodilation, increases flow to non-affected areas of the myocardium, and thus increases the contrast between the normal and abnormal perfused area [29]. Perfusion image abnormalities resulting from the heterogeneity of the coronary blood flow reserve allow clinicians to highlight ischemic areas. The potential interest of the modulation of the adenosinergic system in CAD was first considered and investigated in the 2000s, and the development of potent and selective A_2_AR agonists has been a subject of medicinal chemistry research for the ensuing three decades [30,31,32,33]. The modulation of the adenosine system is commonly used as a vasodilator tool [34,35]. On the other hand, the pharmacological inhibition of the enzymes that catalyze the degradation of adenosine will facilitate adenosine receptor stimulation by increasing the endogenous extracellular adenosine concentration. Thus, the main tracks consist either of increasing the endogenous concentrations of adenosine by decreasing its degradation or inhibiting its reuptake, or of using receptor agonists or antagonists. In this context, adenosine, dipyridamole, and regadenoson are common agents used in pharmacological stress tests.

### 4.1. Use of Adenosine Plasma Level for the Diagnosis of CAD

Accordingly, adenosine is an early and sensitive marker of ischemia, and is released in the extracellular spaces by endothelial cells and myocytes [36]. There have been very few reports on plasma adenosine concentrations in the peripheral blood, because the measurement of adenosine plasma level by high-performance liquid chromatography (HPLC) reported an analytical variability (CVa) ranging from 3% to 10% [37,38,39,40,41,42,43]. High basal adenosine plasma levels were reported in CAD patients, particularly those with severe CAD [24] or in patients with positive exercise stress testing (EST) [26]. In contrast with these studies, CAD was inversely associated with plasma adenosine levels after multivariable adjustment in a large cohort study of >1100 patients undergoing evaluation for acute or chronic CAD [44].

The biological variability influencing circulating adenosine levels should be considered when using adenosine as an end point in clinical studies. This biological variability can be explained by several factors. For example, due to its quick cellular uptake and intracellular metabolism, the half-life of circulating adenosine is counted in seconds [37,38]. To adequately measure the circulating plasma adenosine, the use of a specific stop solution to immediately block its metabolism is required. However, there is no consensus on the use of a specific stop solution [37,38,39,40,41,42,43]. Equally important is the performance of the blood collection for the quality of the results. The speed of blood collection and mixing into the stop solution can change the displayed result [41]. Furthermore, genetic variants in genes encoding enzymes implicated in adenosine metabolism lead to increased endogenous adenosine formation [45,46]. Nevertheless, in this review we report the performance of a high-throughput protocol for rapid HPLC-based adenosine quantification with performance parameters in congruence with good practice guidelines [40,41]. Recently, a highly sensitive ultra-performance liquid chromatography–tandem-mass spectrometry (UPLC-MS/MS) analytical method for plasma adenosine has been developed and validated [47]. This optimized sampling protocol will need adequately powered studies to assess local variability, with future studies requiring robust protocols and statistical methodology. Additionally, and more importantly, the measurement of circulating adenosine does not necessarily reflect the tissue level of adenosine. Therefore, the adenosine concentration will be different among tissues during physiological conditions and may reach very high levels during ischemia. Adenosine is known to accumulate in the extracellular space in response to hypoxia or ischemia [13,33,48]. Saito et al. [49] reported that the local production of adenosine increases during moderate hypoxia in forearm tissue, although this is not reflected in the arterial or venous plasma concentration of adenosine. The local production of adenosine would not be reflected in the artery, owing to a dilution effect. One cannot rule out that higher adenosine tissue levels or greater sensitivity toward the adenosine signaling pathway may still account, at least partially, for the occurrence of the significant benefits of adenosine on the cardiovascular system. The limitation is to assess the local adenosine levels due to the difficulties in technical factors and variability described above.

Since there is considerable biological variability in the adenosine plasma levels and many factors influence the circulating adenosine levels, the interpretation of future studies in humans requires powerful future evaluations of adenosine plasma levels as a marker of coronary artery disease. Furthermore, a lot of inflammatory diseases are associated with adenosine plasma level abnormalities. The lack of specificity makes it unlikely that this marker will be used alone.

### 4.2. Use of Intravenous Adenosine in Stress Imaging for the Diagnosis of CAD

Adenosine infusion acts on the activation of A_2_ receptors and causes coronary vasodilation. It creates a disparity in blood flow between normal and stenosed arteries, leading to ischaemia imaging [34]. Adenosine has several side effects that correlate with the activation of other receptors, such as A_1_R, A_2B_R, and A_3_R. These side effects are hypotension, atrio-ventricular block, bronchospasm, peripheral vasodilatation, and gastrointestinal symptoms [13]. Since adenosine is considered as a potent bronchoconstrictor, this may explain why it is avoided in patients with asthma or chronic obstructive pulmonary disease.

### 4.3. Use of Intracoronary Adenosine for the Diagnosis of CAD

In 1990, Wilson et al. [50] demonstrated in humans that intracoronary adenosine administration leads to maximal coronary vasodilation. Knock out animal models show that the bolus administration of adenosine leads to a significant increase in the coronary blood flow, and that this increase occurs with the highest change two minutes after injection. The increase in coronary blood flow is secondary to the activation of the A_2A_ and A_2B_ receptors, while the decrease in cardiac output is mediated mainly through the A_2B_ and A_3_ receptors [18]. Based on this data, the concept of a myocardial fractional flow reserve has been developed as an invasively determined index of the functional severity of coronary stenosis. At present, invasive coronary angiography is considered the gold standard for the diagnosis of significant CAD [51], but the assessment of coronary artery disease severity by invasive coronary angiography (ICA) is flawed because the angiographic severity of a given epicardial stenosis does not necessarily commensurate with its functional significance. Fractional flow reserve is a standardized and well-established method frequently used in clinical practice to evaluate the hemodynamic significance of epicardial coronary stenosis identified by invasive coronary angiography. It is based on the change in the pressure gradient across the stenosis after the achievement of maximal hyperemia of the coronary circulation, which is commonly induced by the intracoronary administration of adenosine. Maximal hyperemia is a critical prerequisite to correctly assess the fractional flow reserve (FFR) because suboptimal microcirculatory vasodilation might result in the underestimation of the functional severity of coronary stenosis [52]. When the maximum flow in the stenotic artery is normalized by the intracoronary administration of adenosine (50 to 100 µg) in a quick bolus, the effects of pressure, heart rate, or vasomotor tone are reversed [53].

### 4.4. Increasing Endogenous Adenosine in Stress Imaging for the Diagnosis of CAD

Another vasodilator agent usable for pharmacological stress tests is dipyridamole [35]. Dipyridamole inhibits the uptake and metabolism of adenosine, increasing coronary vasodilation and the coronary blood flow (CBF) via the A_2A_ receptors. This agent causes ischemia almost exclusively by altering coronary hemodynamics in the presence of critical epicardial coronary artery disease. The intravenous infusion of dipyridamole is an acceptable alternative to exercise stress for detecting physiologically significant coronary artery stenosis. Since this agent also serves as non-selective adenosine receptor agonists, it also may cause bronchoconstriction and negative chronotropic, inotropic, and dromotropic effects via the A_1_ receptor activation.

### 4.5. Use of Adenosine Receptor Agonists (Regadenoson) in Stress Imaging for the Diagnosis of CAD

Among the selective adenosine agonists used in myocardial perfusion imaging [54], one of these, regadenoson, is a stress agent that selectively activates the A_2A_ receptor, leading to coronary vasodilation. The hyperemic response to regadenoson increases the coronary blood flow with greater intensity in areas perfused by normal arteries, which helps in identifying the ischemic area. Because it is a selective A_2A_R agonist, fewer undesirable side effects of A_1_, A_2B_, and A_3_ receptor stimulation have been noticed. This agent is easier to use, but because the cost of regadenoson is greater than that of other vasodilating agents, it is reserved for patients with asthma and chronic obstructive pulmonary disease.

## 5. Modulating the Adenosinergic System: An Expected Tool for the Treatment of CAD Patients

### 5.1. Use of Intravenous or Intracoronary Adenosine for Therapeutic Intervention

It was established that the administration of adenosine prior to index ischemia reduces the myocardial infarct size in an ischemia/reperfusion animal model [55], and the protective effects of exogenous adenosine on the ischemic myocardium have also been reported in humans [56,57,58,59,60,61,62]. In the prospective, open-label, randomized study named “The Acute Myocardial Infarction STudy ADenosine The Acute Myocardial Infarction STudy Adenosine” (AMISTAD) of patients treated with thrombolysis for ST-elevation myocardial infarction (STEMI), the intravenous administration of adenosine failed to reduce the infarct size and improve clinical outcomes [56]. Only in a post hoc analysis in AMISTAD 2 [57] was adenosine found to reduce infarct size and improve clinical outcomes in a subgroup of patients with the early initiation of reperfusion. Furthermore, the intracoronary administration of adenosine did not increase myocardial salvage in patients treated with percutaneous coronary intervention for ST-elevation myocardial infarction [58]. Several other underpowered studies with small sample sizes [59,60] showed lower levels of clinical outcomes in the adenosine group (risk reduced by greater than 3%), suggesting possible myocardial preservation. A double-blind study of 201 patients with primary percutaneous coronary intervention for STEMI failed to show an effect of adenosine on infarct size [61]. A meta-analysis showed that intracoronary adenosine administered as an adjunct decreases heart failure incidence in STEMI patients [62]. Therefore, studies in humans have not been as promising as experimental reports. These disappointing findings may result in poor intravascular adenosine interstitial delivery (short half-life of adenosine, strong endothelial barrier that empowers exogenous adenosine to reach the cardiomyocyte), despite the high dose of intracoronary administration of adenosine [63].

### 5.2. Increasing Endogenous Adenosine for Therapeutic Intervention

As mentioned previously, several drugs can increase the extracellular endogenous concentration that potentially overcomes the limitations of exogenous adenosine administration. For example, dipyridamole, by inhibiting adenosine uptake by erythrocytes (and thus increasing endogenous adenosine), significantly improves the adenosine cardioprotective effect in experimental studies [64]. To the best of our knowledge, dipyridamole has never been tested as a therapeutic tool in patients with ST-elevation myocardial infarction. Acadesine, a first-in-class adenosine-regulating agent, increases the intracellular and extracellular concentrations of adenosine by competing for the nucleoside transporter and inhibiting adenosine deaminase, respectively. A meta-analysis of previous clinical trials of acadesine [65] showed that this drug might beneficially reduce adverse outcomes after coronary artery bypass grafting (CABG) surgery. In 2012, in a randomized controlled trial (RED-CABG), the effect of the adenosine-regulating agent acadesine was assessed on morbidity and mortality associated with coronary artery bypass grafting. This clinical trial was stopped because the results of a prespecified futility analysis indicated a very low likelihood of a statistically significant efficacious outcome [66].

### 5.3. Use of Adenosine Receptor Agonists for Therapeutic Intervention

Agonists that target the A_1_R, A_2A_R, A_2B_R and A_3_R adenosine receptors have the potential to be potent treatment options for a number of cardiovascular diseases. Because each of these adenosine receptors plays a distinct role throughout the body in different tissue types (from low to high expression), obtaining highly specific receptor agonists is essential. Two selective A_2A_R agonists protected tissue during ischemia-reperfusion injury by reducing the myocardial infarct size without elevating the coronary blood flow or other hemodynamic effects [67,68]. In 2017, Da Silva et al. [69] showed that the oral administration of an A_2A_R agonist (LASSBio-294) in male spontaneously hypertensive rats after experimental-induced myocardial infraction confirmed that this A_2A_R agonist might be an alternative treatment for heart failure due to ischemia.

Unfortunately, the administration of the mixed A_1_, R, and A_2_R agonist (AMP579) did not significantly affect infarct size in patients with STEMI treated with primary percutaneous transluminal coronary angioplasty [70]. Here, again, studies in humans have not been as promising as experimental studies.

### 5.4. Use of Adenosine and Its Receptors as a Potent Inhibitor of Platelet Aggregation

Besides increasing coronary blood flow, in in vitro and in vivo experiments adenosine has been described as a potent inhibitor of platelet aggregation through the A_2_-Rs of the platelets [71]. Only A_2A_Rs and A_2B_Rs are expressed in platelets, but A_2A_Rs play a major role in inhibiting platelet function due to a higher affinity for adenosine [72]. The activation of the A_2A_Rs in platelets has been demonstrated to inhibit platelet aggregation through the elevation of cAMP [73]. The agonistic action of platelet A_2A_Rs might result in an enhanced intracellular cAMP level and consequently lead to the inhibition of platelet activation and aggregation, and thus contribute to the potential for A_2A_Rs-based anti-platelet therapies. This simplified explanation is complicated by the fact that human platelets are not a homogeneous population with size and content variations and by the differential expression of distinct purinergic receptors based on size and function [74]. Recently, Boncler et al. showed that A_2A_Rs agonists used alone and in a combination with P2Y12 antagonists are effective in the enhancement of the inhibition of platelet function without cytotoxicity [75]. Given the multi-factorial roles of platelets and the fact that in vitro studies are more reductionist but of mechanistic interests, the importance of platelet adenosine receptors as therapeutic targets in vivo studies will provide more clinically relevant information regarding their potential benefits therapeutic in patient cardioprotection.

## 6. Pharmacological Profile of A_2A_ Receptors in CAD Patients

### 6.1. Use of Spare A_2_Rs Detection for the Diagnosis of Inducible Myocardial Ischemia in CAD Patients

The expression levels of adenosine A_2_Rs and their functional activity are therefore of paramount importance in coronary blood flow maintenance. In a previous review, Fenouillet et al. [76] described the concept of spare receptors. The receptor reserve concept was originally defined as the fraction of receptors not required to achieve maximal response by a full agonist [77]. The presence of spare receptors seems to be an adaptive mechanism to counterbalance a low level of agonist or a low level of receptors expression of both [76]. The spare receptor model can be designed as a signal amplification system in which the effectiveness of the response to different ligands, full or partial agonists, can be quite complex due to the dissociation between receptor activation level and biological effects. The presence of spare receptors is suspected when a maximal cAMP production occurs, while only a weak fraction of receptors are activated by the ligand. In CAD patients, a low A_2A_R level, associated with a maximal level of cAMP production, which is consistent with the presence of spare receptors, is estimated to be associated with significant ischemia. The association between spare receptors and inducible myocardial ischemia was found in CAD patients with a positive exercise stress test in those with a positive fractional flow reserve and in patients with lower extremity peripheral artery disease, which is associated with severe CAD [26,78,79].

### 6.2. Use of Extracellular Vesicles with Ubiquitinated Adenosine A_2A_ Receptor in Plasma for the Diagnosis of CAD

The adenosine A_2A_R expression measurement procedure for spare determination is not easy because of its need to use the agonist that binds to adenosine receptors in an “irreversible manner” (irreversible meaning at least during the experimental procedure). Another simpler possibility would be the detection and measurement of A_2A_R in an exosomal fraction. Exosomes are extracellular vesicles of endosomal origin that have emerged as key mediators of intercellular communication and represent one of the most intensely studied and rapidly growing areas of research. Ruf et al. [80] searched for the presence of A_2A_R in extracellular vesicles from the blood of CAD patients compared with controls (healthy individuals). The results show that plasma from CAD patients contains extracellular vesicles carrying ubiquitinated A_2A_R. Ubiquitin might function here as a carrier for A_2A_R delivery into the blood, likely intended to be transported to other tissues or to be eliminated. Because there is a correlation between the homocysteine plasma level and the A_2A_R concentration in exosomes of CAD patients, it is suspected that homcysteine may play a role in the decrease in A_2A_R expression on peripheral blood mononuclear cells of CAD patients on the one hand and in the incorporation of A_2A_R in exosomes on the other hand. However these results need to be confirmed in a great number of patients, in a multi centric study approach. Finally, the precise link between plasma high homocysteine levels and A_2A_R incorporation in exosomes remains unknown. From a clinical point of view, extracellular vesicles with A_2A_R constitute a potential diagnostic tool in coronary artery disease by a simple, rapid, and reliable solvent-based protein precipitation method, as previously described. In summary, the A_2A_R expression level from a single blood sample may help to screen for significant coronary artery disease using peripheral blood mononuclear cells. Let us speculate that in the future, the level of exosomes in relation to the A_2A_R expression level, released by exocytosis into the extracellular space and representing a subtype of extracellular vesicles derived from endosomes, will be used to screen for CAD membrane particles or vesicles in a right-side-out or inside-out configuration, which could also be released in the blood during coronary artery disease.

## 7. Conclusions

This review focused on the adenosine A_2A_ receptor subtype, circulating adenosine levels, and their relationship with coronary artery disease. The disappointing findings regarding adenosine plasma levels in the field of coronary artery disease could be explained by a high rate of variability and other factors yet to be determined. Furthermore, the variability does not necessarily reflect the adenosine tissue level. Following the discovery of the importance of adenosine and its receptors in the cardiovascular system, the evaluation of the adenosinergic profile (i.e., adenosine level and receptor expression) and the use of adenosine agonists has become more prevalent and makes the A_2A_R a fascinating target for CAD screening. On the basis of a large number of publications in the literature, A_2A_R ligands have potential roles in a wide spectrum of cardiovascular disease conditions, such as therapeutic and diagnostic conditions. The production of highly specific agonists to each receptor remains a slow and challenging process. This also helps us to understand in depth the essential and specific roles mediated by adenosine receptor subtypes throughout the body. Some of these ligands have been patented and are currently undergoing clinical evaluation for different therapeutic applications.

## Figures and Tables

**Figure 1 ijms-21-05321-f001:**
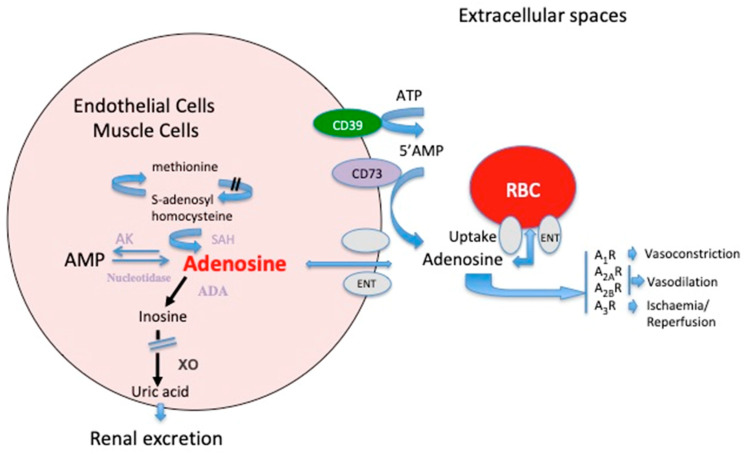
Schematic representation of the adenosine metabolism and its effects on vessels. At the extra cellular level, the main source of adenosine is the dephosphorylation of ATP and AMP via the ectonucleotidases CD39 and CD73. In the intracellular spaces, adenosine comes from AMP dephosphorylation though cytosolic nucleotidase. Part of adenosine comes from the methionine cycle. ADA: adenosine deaminase; AK: adenosine kinase; XO: xanthine oxidase; SAH: s-adenosyl homocysteine hydrolase; A_1_R, A_2A_R, A_2B_R, and A_3_R are adenosine receptor subtypes; ENT: equilibrative nucleoside transporter.

**Figure 2 ijms-21-05321-f002:**
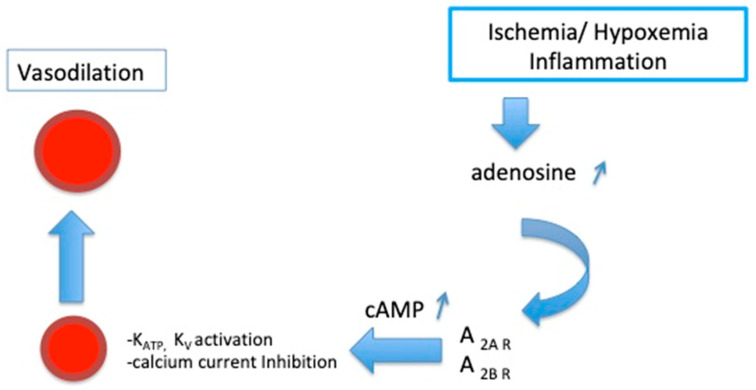
Schematic representation of the activation of A_2_ receptor subtypes on coronary arteries. K_ATP_: ATP-sensitive potassium channels; K_V_: voltage-sensitive potassium channels.

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
