# Peer review of "Adenosine and Its Receptors: An Expected Tool for the Diagnosis and Treatment of Coronary Artery and Ischemic Heart Diseases"

_ijms, 2020, doi:10.3390/ijms21155321_

Round 1

Reviewer 1 Report

Adenosine and its receptors: an expected tool for the 2 diagnosis and treatment in coronary artery and 3 ischemic heart diseases by Marine Gaudry et al. The paper by Gaudry et al. concerns adenosine receptors and their role in the treatment of CAD and IHD. In general, it is a nicely written paper, with the main focus on the modulation of endogenous adenosine concentrations in triggering the effects desirable in CAD patients. It is a very brief review, with 7-8 pages of essential text, co-authored by 14 authors. It may seem a little bit striking that rather short review has so many authors, however, it is not a critical remark. The title may be a little too promising, particularly with the respect to the treatment aspect. The paper essentially concerns standard treatment with adenosine, the concept popular in early 2000s. In my opinion, a too little attention was paid to combine and discuss more profoundly the hypotensive and antithrombotic effects. I also miss some discussion of the possible effects of adenosine and its derivatives on blood platelets and endothelial cells and the role of their interplay in CAD patients’ treatment. The authors make a promise in the beginning of the papers to discuss various adenosine derivatives on adenosine receptors, but they merely mention one pharmaceutical – regadenoson. There is quite an interesting literature on various adenosine and non-adenosine derivative agonists of AR and it would be worthy to write more about it in a review on adenosine receptors. There is also a recently developed approach to use a dual therapy and combining classical P2Y12 antagonists and AR agonists, which may appear promising. I would urge the authors to discuss this idea more profoundly in their review. In general, the [aper is interesting and worth of considering upon referring of the authors to the above-mentioned points

Author Response

We would like to thank reviewer 1 for describing very precisely the errors and mismatch of references in this manuscript. We have reorganized the reference and check the match reference carefully. For example, we found and corrected another discrepancy concerning references 61, 62, 63.  

We will respond to the various relevant remarks and the above-mentioned points."

Question 1

Thank you very much for pointing out the concerns the shortcomings of the references.

Indeed reference 13 was spread over 3 numbers 13, 14, 15 which caused a shift in the numbering of the references. We have therefore corrected this for reference 13

Question 2

Indeed, the references 36, 37, 38, 54, 57, 58, 65, 66, 67, 68, 70, 71, 72, and 73 have two different numbers.

we have corrected this anomaly

Question 3 and 4

Saito is now attributed to reference 49 and reference 50 to Wilson.

Reviewer 2 Report

This manuscript by Gaudry and colleagues review Adenosine and its receptors for the diagnosis and treatment in the coronary artery and ischemic heart diseases. Adenosine is an endogenous nucleoside and released mostly by endothelial cells and myocytes during ischemia or hypoxia and regulates the cardiovascular system via four specific G-protein‐coupled receptors, named A1 R, A2AR, A2BR and A3R. This review aims to focus on the specific exploration of the adenosine plasma level and its relationship with the A2A receptor, which seems a promising biomarker for a diagnostic and/or a therapeutic tool for the screening and management of coronary artery disease. This is a review article and needs to collect a lot of references to overview the current knowledge to focus on specific survey. However, the organization of references by the authors is not correct and several references in the article are miss or in the wrong order that could be misleading to cite the wrong reference. The following comments summarize my concerns:
1. In line 368, the reference 13th is incomplete, as well as reference 14, 15 are missing.
2. In the line 421, 425, 426, 473, 479, 482 510, 513, 517, 520, 527, 531, 535 and 545, the references 36, 37, 38, 54, 57, 58, 65, 66, 67, 68, 70, 71, 72, and 73 have two differ code.
3. in line 163, on page 5, The authors state that reference 49 was cited from Saito et al, however when I look at reference 49 on page 11, line 469, that is Lofgren L et al. Saito et al, is the reference 51.
4.also the same problem, in line 185, at page 5, The authors state that the reference 50 was cited from wilson et al, but when I look at the reference 50 in page 11, line 469, that is Kiers D et al. wilson et al, is the reference 52.
The biggest problem in this review article is a mismatch of the reference in the manuscript. The order of references in the article is mismatched within the reference section from page 8 to 13.
The authors need to reorganize the reference and check the match reference be careful.

Author Response

We would like to thank reviewers 2 for the general appreciation of this review qualified

“a nicely written paper interesting and worth of considering “ and we will respond to the various relevant remarks and the above-mentioned points."

Regarding the rather short review with so many authors, the length of the text has been imposed by the publisher. The recommendations to the authors stipulate a maximum of 4000 words including the reference.

The contribution of each author has been specified by an addendum which has been added.

As mentioned by the reviewer, only pharmaceutical-regadenoson was discussed in our review, we add this sentence to refer to the 54 reference, line 219 “Among selective adenosine agonists used in the myocardial perfusion imaging [54, see for a review], one of this, » for the readers interesting in this subject.

We thank reviewer 2 for pointing out the lack of precision concerning the potential role of A2R receptors in the field of thrombosis and particularly their interaction with P2Y12       antagonists agents. Our title refers to the role in the diagnosis and treatment of coronary artery disease. The antithrombotic role of adenosine, A2R receptors and its agonists and/or antagonists needed to be clarified. We have therefore added a new paragraph to this effect (ligne 277 paragraph 5.4) named 5.4. Use of adenosine and its receptors as a potent inhibitor of platelet aggregation). It is now exist a discussion of the possible effects of adenosinergic system on blood platelets and endothelial cells and the role of their interplay in the treatment of CAD patients. Indeed a developed approach to use a dual therapy and combining classical P2Y12 antagonists and AR agonists ,in order to reduce the undesirable sides effects (bleeding) or limit the high on treatment platelet reactivity (HTPR) so called   resistance to clopidogrel. These works has been described and listed in the reference (Ref 71-75).

Round 2

Reviewer 2 Report

No further questions 

This manuscript is a resubmission of an earlier submission. The following is a list of the peer review reports and author responses from that submission.

Round 1

Reviewer 1 Report

The authors summarized the role of adenosine and a2a receptors in terms of diagnostic and therapeutic potentials mainly in human disease. The manuscript appears to be unbalanced and favor to clinical evidence without revealing the overall experimental data.

As the authors indicated in their clinical evidence with outcomes that are not as promising as experimental data, the title sounds too promising and needs to be neutralized to a more objective one.

In many places in the manuscript, authors focus too much about the expression of A2A receptors as very abundant one in the coronary vasculature without specifying which species/cell types are they talking about. In swine model, many studies including a recent one demonstrated a more abundant A2B but not A2A in coronary vessel/SMCs (Sun et al. J Pharmacol Sci 2019) despite a prominent function for A2A but not A2B. More literature work needs to be done for the authors to have a in depth discussion on receptor expression vs. receptor sensitivity, if they would like to include expression evidence in the review article. Another point is that in patients with coronary artery disease exhibits low A2A expression (line 153), of which the authors should not conclude that the low expression is associated with poorer coronary function. Since they never measured the expression in healthy condition. Again, evidence on healthy coronary microcirculation in animal models is valuable.

When talking about the plasma level of adenosine, are there any evidence for the local circulating adenosine levels in coronary microcirculation?

Authors mentioned other A2A agonists : binodenoson, sonedenoson and apadenoson. How about studies for these drugs?

The exosome carrying A2A is quite interesting. Is there more information in the literature? How about the level of A2A in exosomes from healthy subjects?

Some figures are necessary to illustrate several sections.

Double check the "vasodilation" vs "vasodilatation" as well as "A2AR" vs "A2AR"

Author Response

The authors summarized the role of adenosine and a2a receptors in terms of diagnostic and therapeutic potentials mainly in human disease. The manuscript appears to be unbalanced and favor to clinical evidence without revealing the overall experimental data.

As the authors indicated in their clinical evidence with outcomes that are not as promising as experimental data, the title sounds too promising and needs to be neutralized to a more objective one.

In many places in the manuscript, authors focus too much about the expression of A2A receptors as very abundant one in the coronary vasculature without specifying which species/cell types are they talking about. In swine model, many studies including a recent one demonstrated a more abundant A2B but not A2A in coronary vessel/SMCs (Sun et al. J Pharmacol Sci 2019) despite a prominent function for A2A but not A2B. More literature work needs to be done for the authors to have a in depth discussion on receptor expression vs. receptor sensitivity, if they would like to include expression evidence in the review article.

Answer : Sun et al is now cited in 1 and 4. The importance of sensitivity vs expression is now precised (second paragraph of 4).

Another point is that in patients with coronary artery disease exhibits low A2A expression (line 153), of which the authors should not conclude that the low expression is associated with poorer coronary function. Since they never measured the expression in healthy condition. Again, evidence on healthy coronary microcirculation in animal models is valuable.

Answer : A2A R expression was not evaluated in coronary arteries of healthy subject for ethical consideration. However, it was reported that A2A R expression and function evaluated in coronary tissues (see ref 23 Gariboldi et al), and more generally in the cardiovascular system (Ref 30 Varani et al ) mirrors those found on PBMC. In this context, it was reported a lower expression on A2A R in CAD patients vs healthy subjects

When talking about the plasma level of adenosine, are there any evidence for the local circulating adenosine levels in coronary microcirculation?

Answer : you are wright, we have to date no information on adenosine concentration in microcirculation

Authors mentioned other A2A agonists : binodenoson, sonedenoson and apadenoson. How about studies for these drugs?

Answer : These drugs are poorly used probably because of a lack of specificity and price.

This is now specified at the end of the corresponding paragraph 5.2.4

The exosome carrying A2A is quite interesting. Is there more information in the literature? How about the level of A2A in exosomes from healthy subjects?

Answer : to the best of our knowledge there Ruf et al cited in reference number 42, is the only one. It was reported a very low if any A2A R level in exosomes of healthy subjects. However only severe CAD exhibit A2A R in exosomes. Furthemore it was described a correlation between homocysteine level and A2A R in exosomes but on a very low number of patients that need confirmation. This is now specified in the croreesponding paragraph 5.2.7

Some figures are necessary to illustrate several sections.

Answer : Two figures have been added. Figure 1 is a schematic representation of adenosine metabolism and figure 2 is a schematic representation of the action of adenosine on coronary arteries

Double check the "vasodilation" vs "vasodilatation" as well as "A2AR" vs "A2AR"

Answer : this was corrected

Reviewer 2 Report

The authors made an attempt to review the state of the art of utilization of adenosine receptors in  diagnostics and therapy of heart ischemia. Since there is a scarcity of papers reviewing this subject in a recent dozen of years, there is a need of such a review. The authors first present general information on adenosine generation and its role in physiological conditions and they continue by showing how this nucleoside and its receptors can serve diagnosing or treating of ischemic heart disease. The general concept of the review is accurate, but numerous flaws make the paper hard to read and they cause that the paper is much less valuable source of information than it could have been.

Some fragments of the text do not fit to the lineage or have misleading titles: chapter 3. which is entitled  ‘Circulating adenosine’ in its most part deals with the methods of assaying of circulating adenosine and not of the sources of the nucleoside etc. Chapter 5.2.8 describes in much details the procedure of assessment of ischemia, which takes nearly half of the page, and relation of this part to adenosine is rather marginal.

There are several instances where the sentences are repeated throughout the text: the sentence in line 43 starting with: “The effects of adenosine on…” is repeated in line 45. The sentence in line 113 starting with “Therefore, the adenosine…” is repeated in the very next sentence in line 115”(…) and the concentration of the nucleoside…”. The sentence in line 242 starting with “The stress test…” is repeated in line 271. Also entire chapters seem to duplicate itself. For instance chapter number 5.2.3 actually replicates the preceding chapter 5.2.2. They even have almost the same titles. Entire chapter 5.2.8 again describes the use of adenosine infusion for CAD diagnosis. Such obvious duplications are not only an evidence of a certain lack of focus during writing of the manuscript but also give an impression that no one had really read the paper before its submission.

There are numerous examples of inaccurate usage of biological terms or mistakes: there is no such thing as “coronary cells” (lines 26 and 137), authors probably meant coronary endothelial cells; Gq protein does not inhibit adenylyl cyclase (line 131), it activates phospholipase C instead; proteins are rather not expressed in human red blood cells (line 58); why do authors think that synthesis of AR agonists requires “expensive and hazardous radioactive agents”? (line 218), adenosine receptors by no means produce cAMP (lines 153-154).

Numerous sentences are awkward: Line 57: adenosine deaminase is according to the authors: ubiquitous, synthetized by the liver, present in most cells and expressed particularly in RBCs. So where it is not present? Line 94: “patients on ticagrelor therapy or not”. What can this mean? Line 172: “(…) diagnostic applications will be discussed via exogenous or increased release (…)”. How can be something discussed by release? Line 179: “myocardial infarction infarct size”. Line 337: a word “cohort” used 3 times in the same sentence.

Lots of abbreviations or terms are not explained: “STOP solution” (line 73) – is it simply “stop solution” or STOP stands for something else; “AMPD  1” (line 77), “index ischemia” (line 178); ADO (line 161), PBMCs (line 147).

Is URIC acid in line 127 simply a uric acid?

Chapters are written in non-ordered, unfocused way. A good example of it is the chapter number 5.2.7. In line 292 the authors state that “The adenosine A2AR expression procedure is meticulous “, suggesting the expression of a recombinant protein. But the authors probably mean assay of the level of protein expression. The whole chapter (5.2.7) seems to be upside-down. Its title suggests that it will explain why A2A in EV can be used to diagnose CAD, but it starts with procedure of assay going down to details such as antibodies concentrations. The very idea stated in the title is roughly explained in the end of the chapter. What is more, although it is extracellular vesicles what is supposed to be used to assay of A2A, at the same time in line 304 it says that “peripheral blood mononuclear cells”  could be used for the assay. What is the relation of the two?

Why references are used in chapters’ titles? (lines 248, 258, 267, 290)

Reference 31 in line 139 is not correct. The cited paper is not related to cAMP production.

The manuscript contains a lot of important information and could it be a source of valuable information but it must be profoundly rewritten.

Author Response

The authors made an attempt to review the state of the art of utilization of adenosine receptors in  diagnostics and therapy of heart ischemia. Since there is a scarcity of papers reviewing this subject in a recent dozen of years, there is a need of such a review. The authors first present general information on adenosine generation and its role in physiological conditions and they continue by showing how this nucleoside and its receptors can serve diagnosing or treating of ischemic heart disease. The general concept of the review is accurate, but numerous flaws make the paper hard to read and they cause that the paper is much less valuable source of information than it could have been.

Some fragments of the text do not fit to the lineage or have misleading titles: chapter 3. which is entitled  ‘Circulating adenosine’ in its most part deals with the methods of assaying of circulating adenosine and not of the sources of the nucleoside etc.

Answer ; the title of the Chapter is now : The importance of circulating adenosine evaluation

Chapter 5.2.8 describes in much details the procedure of assessment of ischemia, which takes nearly half of the page, and relation of this part to adenosine is rather marginal.

There are several instances where the sentences are repeated throughout the text: the sentence in line 43 starting with: “The effects of adenosine on…” is repeated in line 45. The sentence in line 113 starting with “Therefore, the adenosine…” is repeated in the very next sentence in line 115”(…) and the concentration of the nucleoside…”. The sentence in line 242 starting with “The stress test…” is repeated in line 271. Also entire chapters seem to duplicate itself. For instance chapter number 5.2.3 actually replicates the preceding chapter 5.2.2. They even have almost the same titles. Entire chapter 5.2.8 again describes the use of adenosine infusion for CAD diagnosis. Such obvious duplications are not only an evidence of a certain lack of focus during writing of the manuscript but also give an impression that no one had really read the paper before its submission.

Answer ; This part has been comletely rewriten and shortened ; 5.2.2 and 5 .2.3 have been merged

There are numerous examples of inaccurate usage of biological terms or mistakes: there is no such thing as “coronary cells” (lines 26 and 137), authors probably meant coronary endothelial cells; Gq protein does not inhibit adenylyl cyclase (line 131), it activates phospholipase C instead; proteins are rather not expressed in human red blood cells (line 58); why do authors think that synthesis of AR agonists requires “expensive and hazardous radioactive agents”? (line 218), adenosine receptors by no means produce cAMP (lines 153-154).

Answer ; these mistakes have been corrected

Numerous sentences are awkward: Line 57: adenosine deaminase is according to the authors: ubiquitous, synthetized by the liver, present in most cells and expressed particularly in RBCs. So where it is not present?

Answer : the sentence has been rewritten

Line 94: “patients on ticagrelor therapy or not”. What can this mean?

Answer ; this sentence has been suppressed

Line 172: “(…) diagnostic applications will be discussed via exogenous or increased release (…)”. How can be something discussed by release? Line 179: “myocardial infarction infarct size”. Line 337: a word “cohort” used 3 times in the same sentence.

Answer : the sentence has been corrected

Lots of abbreviations or terms are not explained: “STOP solution” (line 73) – is it simply “stop solution” or STOP stands for something else; “AMPD  1” (line 77), “index ischemia” (line 178); ADO (line 161), PBMCs (line 147).

Is URIC acid in line 127 simply a uric acid?

Answer : this has been corrected

Round 2

Reviewer 1 Report

Although authors have partially revised the manuscript and provided two new figures, the first question was not addressed sufficiently. This is basically lack of balanced experimental evidence, and the request to revise the title to fit the less promising clinical outcomes by adenosine treatment was ignored.

Reviewer 2 Report

The manuscript has been amended to some extent. Some of my remarks were taken into account but the other ignored: The sentence in line 242 starting with “The stress test…” is repeated in line 265. Despite authors’ claims abbreviation PBMC has not been explained.

Line 162 “A2AR from patients with CAD is poorly expressed and, consequently, produces low levels of cAMP” Again, A2AR do not produce cAMP. I asked to correct this in the first round. Authors claim in their response that it has been corrected but apparently has not.

Chapter 5.2.2 and 5.2.8 are basically about the same and I don’t understand why they are separated.

Still the relation of exosomes to PBMC has not been explained in chapter 5.2.7: title of the chapter says that extracellular vesicles are used for the assays but in the very first sentence the mysterious PBMC pop-up. What is more the sentence starts with “The adenosine A2AR expression measurement procedure is long and difficult” as if the described procedure was the only procedure to assay A2AR expression. Have the authors heard about flow cytometry?

Chapter 3 deals with technical aspects of adenosine assays and it is pasted between two chapters dealing with physiological aspects of adenosine. It does not fit here. It should moved and shortened.

Line 128 “Extracellularly, adenosine activated four distinct receptors named” it should rather be “activates”

Line 131 “are positively coupled to Gs proteins” can Gs protein be negatively coupled to the receptors?

Line 136, 154 what are “coronary tissues”?

Authors merged chapters 5.2.2 and 5 .2.3 but they did not bother to change the numeration: 5.2.2 is followed by 5.2.4

The title “Adenosine and Its Adenosine A2 Receptors” should rather be replaced with Adenosine and its receptors….”

In its first version the manuscript made an impression of a patchwork composed of parts written by several authors and not really merged to one entity. In its amended version the manuscript looks the same. In my first round of the review I suggested that the paper has to be profoundly rewritten. It has not been. In several instances the corrections I suggested were ignored despite the authors claims that they have been corrected. I can’t understand how is this possible with fourteen authors involved.